Optimal selection of COVID-19 vaccination sites in the Philippines at the municipal level

Cabanilla Kurt Izak
Enriquez Erika Antonette T.
Velasco Arrianne Crystal
Mendoza Victoria May P. vmpaguio@math.upd.edu.ph
http://orcid.org/0000-0003-3507-0327 Mendoza Renier rmendoza@math.upd.edu.ph
Institute of Mathematics, University of the Philippines Diliman , Quezon City , Philippines
Arifin Hidayat
Electronic publication date: 2022 Sep 30
Publication date: 2022
Volume: 10
Electronic Location ID: e14151
Received 2022 Jun 21; Accepted 2022 Sep 7
Copyright: © 2022 Cabanilla et al.
Copyright year: 2022
Copyright holder: Cabanilla et al.
License: This is an open access article distributed under the terms of the Creative Commons Attribution License, which permits unrestricted use, distribution, reproduction and adaptation in any medium and for any purpose provided that it is properly attributed. For attribution, the original author(s), title, publication source (PeerJ) and either DOI or URL of the article must be cited.
License URL: https://creativecommons.org/licenses/by/4.0/

Keywords: COVID-19, Dynamic programming, Facility location, Genetic algorithm, Multi-objective optimization, Open street maps, Philippines, Vaccination

Funding: Grant for research on COVID-19 in the Philippines This work is funded by the project titled “Funding for the establishment of a computational research laboratory in the University of the Philippines Diliman Institute of Mathematics, pursuant to section 10(x) of Republic Act No. 11494” under the category “Grant for research on COVID-19 in the Philippines”. The funders had no role in study design, data collection and analysis, decision to publish, or preparation of the manuscript.

==============================
In this work, we present an approach to determine the optimal location of coronavirus disease 2019 (COVID-19) vaccination sites at the municipal level. We assume that each municipality is subdivided into smaller administrative units, which we refer to as barangays. The proposed method solves a minimization problem arising from a facility location problem, which is formulated based on the proximity of the vaccination sites to the barangays, the number of COVID-19 cases, and the population densities of the barangays. These objectives are formulated as a single optimization problem. As an alternative decision support tool, we develop a bi-objective optimization problem that considers distance and population coverage. Lastly, we propose a dynamic optimization approach that recalculates the optimal vaccination sites to account for the changes in the population of the barangays that have completed their vaccination program. A numerical scheme that solves the optimization problems is presented and the detailed description of the algorithms, which are coded in Python and MATLAB, are uploaded to a public repository. As an illustration, we apply our method to determine the optimal location of vaccination sites in San Juan, a municipality in the province of Batangas, in the Philippines. We hope that this study may guide the local government units in coming up with strategic and accessible plans for vaccine administration.

Introduction

The coronavirus disease 2019 (COVID-19), which was first reported in Wuhan, China, has spread across the globe and was declared a pandemic by the World Health Organization (WHO) on March 11, 2020 (Xu et al., 2020; Cao, 2020). Initial findings suggest that vaccination can protect the population against infection (Amit et al., 2021; Levine-Tiefenbrun et al., 2021; Thompson et al., 2021) and may reduce onward transmission (Eyre et al., 2022). COVID-19 vaccines have been shown to be safe and can protect against severe disease, hospitalization, and death (WHO, 2022a). An effective vaccination campaign can lessen the probability of disease resurgence and alleviate the economic burden of the pandemic (Ella & Mohan, 2020). With the constant emergence of new variants of the virus and the waning of immunity provided by vaccines or infection, a herd immunity threshold for COVID-19 seems impossible to identify (Aschwanden, 2021; Morens, Folkers & Fauci, 2022). Classical herd immunity threshold is described as the proportion of the population with immunity, induced by vaccine or infection, against a disease wherein above this threshold, transmission is considerably prevented (John & Samuel, 2000; Fine, Eames & Heymann, 2011; Jones & Helmreich, 2020; Randolph & Barreiro, 2020; Morens, Folkers & Fauci, 2022). Nevertheless, both non-pharmaceutical interventions and vaccination of as many people as possible are necessary for optimal control of COVID-19 (Kadkhoda, 2021; Morens, Folkers & Fauci, 2022).

Besides the global shortage of vaccine supply during the early vaccination phase, safety concerns, vaccine brand hesitancy, and misinformation were among the challenges that delayed vaccination in the Philippines (Huh & Dubey, 2021; Amit et al., 2022). The online survey done before the vaccine rollout in the Philippines revealed that around 70% of the respondents would only get vaccines after many other people or politicians have been vaccinated, and about 97% were worried about fake vaccines (Caple et al., 2022). Proper handling and storage, and fast distribution of COVID-19 vaccines posed logistics challenges, particularly the preparedness of cold chain infrastructure at the national and local levels (Park et al., 2021; Reyes, Dee & Ho, 2021). Shortage of healthcare workers who can administer vaccines also hindered the expansion of the rollout (Sales et al., 2022). As of 9 May 2022, around 68% of the Philippine population aged 5 years or older have been vaccinated with the primary doses. However, vaccine coverage among different age groups and regions greatly varies (WHO, 2022b). Most people who have not received a single vaccine dose include those in geographically isolated and disadvantaged areas, as well as around 2.4 million of the elderly population, who do not have affordable and practical resources to go to vaccination sites. Strategies such as house-to-house campaigns to reach these vulnerable groups and encourage getting vaccinated are being done in a few provinces (WHO, 2022c). A collaborative effort among several stakeholders and sectors is therefore needed to address these issues to protect and save more lives (Corpuz, 2021; WHO, 2022c). We hope that this study may guide the Philippine local government units in coming up with more strategic plans for vaccine administration. We present a way to select optimal vaccination sites from already existing facilities to make the vaccines more accessible to the public and accelerate recovery of the nation from this pandemic.

Our proposed approach solves a facility location program, which is a problem that minimizes the cost of satisfying a set of demands with respect to some set of constraints (Facility Location, 2009). Facility location problem has a variety of applications including determining optimal locations of solar power plant sites (Wang et al., 2020), hydrogen production sites (Yee et al., 2020), tsunami sensors (Ferrolino, Lope & Mendoza, 2020; Ferrolino et al., 2020), infrastructure maintenance depot (Kim & Kim, 2021), tower sites for early-warning wildfire detection systems (Heyns et al., 2020), and high-speed train stations (Chanta & Sangsawang, 2020), among others. Facility location has also been used in several COVID-19-related studies. In Buhat et al. (2020), an optimal allocation of COVID-19 testing kits among accredited testing centers has been proposed. The optimal location of pharmacies for COVID-19 testing to ensure access has been studied in Risanger et al. (2021). Identification of locations of COVID-19 emergency logistic centers has been proposed in Wang & Ma (2021). In Taiwo (2020), optimal COVID-19 testing facility sites in Nigeria have been studied.

Several studies have been conducted on the applications of facility location problems in COVID-19 vaccination distribution strategies. In Bertsimas et al. (2021), an approach to optimize vaccine distribution strategies has been proposed by selecting locations that will minimize the death toll. The method relies on an epidemiological model to capture the effects of vaccination against, and mortality caused by COVID-19. In Basciftci, Yu & Shen (2021), a mathematical framework for finding the optimal locations of distribution centers for test kits and vaccines has been developed. In Buhat et al. (2021), a linear programming model was used for COVID-19 vaccine allocation in the Philippines at the national level. The scale of these studies necessitates the consideration of logistic constraints (e.g., shipping cost, production capacity, operating cost, etc.). In this study, we consider a local-scale vaccination strategy. By doing so, we can focus on finding the optimal location of vaccination sites that will make the vaccines more accessible to the population of a municipality. Our work can be used in conjunction with vaccine allocation methods at the national level (Bertsimas et al., 2021; Basciftci, Yu & Shen, 2021; Buhat et al., 2021). Once the vaccines are allocated to a municipality, our method can be applied to identify the sites where the vaccines will be distributed. A vaccine strategy in Germany (Leithäuser et al., 2021) has been proposed that considers three different objectives, including minimizing the sum of travel distances. In their study, the user can choose which objective they intend to prioritize. In our work, a single cost function is proposed to incorporate all the objectives. Alternatively, we also propose a bi-objective optimization approach that considers both distance and population coverage so that the policymaker may choose among multiple optimal solutions a strategy that prioritizes the needs of the municipality. In Zhang et al. (2022) and Tang et al. (2022), the distance between the vaccination site to the recipient was one of the objectives to be minimized. However, these studies use Euclidean distance, which is not realistic in a municipality setting. In this study, we utilize the Open Street Maps (OSM) and its corresponding Python package OSMNX, to calculate the actual road distance.

In the next section, we formulate three mathematical optimization models to address the vaccination site location problem. The first model incorporates the distance to the sites, COVID-19 cases, and population density in a single-objective function. As an alternative decision support tool, we also present a multi-objective optimization model. The third model is a dynamic optimization problem that recalculates the optimal vaccination sites considering the remaining unvaccinated population of the barangays. Then, we discuss the numerical methods and open-source software used to solve the minimization problems. We illustrate how our proposed method works by using the method to identify the optimal vaccination sites in San Juan, Batangas, Philippines. Finally, we present our conclusions and recommendations for future research.

Optimization problem

Our goal is to determine the optimal location of Lvaccination sites in a municipality from a list of M possible vaccination sites. We consider existing facilities such as public schools and hospitals as possible vaccination sites. Furthermore, suppose that the municipality is divided into N administrative units, which we refer to as barangays. These barangays or villages are usually the country’s basic units of government. Let {Vi:i=1,2,…,M} be the set containing the locations of the M possible vaccination sites. Each Vi is represented by a two-dimensional vector whose components are the latitude and longitude of the ith vaccination site. Define {Bj:j=1,2,….N} as the set containing the location of the N barangays. We can set Bj as the location of the barangay hall, which is usually situated at the center of the barangay. Similarly, each Bj is a two-dimensional vector whose components are the latitude and longitude of the jth barangay.

Define d(Vi,Bj) as the distance of the vaccination site Vi from the barangay hall Bj. In facility allocation problems, different distance measures are used. For example, the Euclidean distance was used in Wong et al. (2009). It was argued in Du, Zhang & Xia (2005) that the l1 distance (also known as Manhattan distance) is more accurate in modeling the driving distance in a city road network. However, in rural municipalities, the roads may not follow a rectangular grid pattern. Since Vi and Bj are accessible via the road network of a municipality, we utilize OSMNX (Version 1.2.1) to calculate the actual driving distance from Vi to Bj. This approach makes the computation of distance more realistic.

Now, suppose L=1, that is, only one vaccination site is assigned to the whole municipality. Then, one distribution strategy is to choose the vaccination site that lies the closest to all the Bj′s. That is, we solve

(1) min1≤i≤M⁡∑j=1N⁡d(Vi,Bj).

However, the minimization problem in Eq. (1) does not take into consideration the population of the barangays. To resolve this, we add more weight on the vaccination sites that are closer to the more populous areas of the municipality. Define Pj as the population of the jth barangay and Tp as the total population of the municipality. Then ∑j=1N⁡Pj=Tp and the problem becomes

(2) min1≤i≤M⁡∑j=1NPjTpd(Vi,Bj).

Moreover, we want to place the vaccination sites near barangays with high numbers of confirmed COVID-19 cases. Define Cj as the number of confirmed COVID-19 cases in the jth barangay and Tc as the total number of confirmed COVID-19 cases in the municipality. Then ∑j=1N⁡Cj=Tc. Similar to how population is incorporated in Eq. (2), we add weights on the barangays with high number of confirmed COVID-19 cases. Thus, we solve

(3) min1≤i≤M⁡∑j=1N⁡[PjTp+CjTc]d(Vi,Bj).

Next, we consider the case when the number of vaccination sites is more than one, that is, L≥2. If there are L vaccination sites, we want the resident of the jth barangay to go to the nearest vaccination site. We can generalize the minimization problem in Eq. (3) as follows:

(4) min1≤i1,i2.…,iL≤M⁡∑j=1N⁡[PjTp+CjTc]min⁡{d(Vi1,Bj),d(Vi2,Bj),…,d(ViL,Bj)}.

The formulation in Eq. (4) successfully accounts for the population density, number of confirmed COVID-19 cases, and distances of the barangays to the vaccination sites.

To make the optimization problem a more flexible decision support tool, we can also consider two goals: minimize the total distance from vaccination sites to barangays and

maximize the total population that are within a pre-defined radius ( ε) from the vaccination sites.

Hence, we can redefine the minimization problem in Eq. (4) as the bi-objective optimization

(5) min1≤i1,i2.…,iL≤M⁡[F1(Vi1,Vi2,…,ViL)F2(Vi1,Vi2,…,ViL)]

where

F1(Vi1,Vi2,…,ViL)=∑j=1NCjTcmin⁡{d(Vi1,Bj),d(Vi2,Bj),…,d(ViL,Bj)},

F2(Vi1,Vi2,…,ViL)=−∑i=1L⁡∑j∈A⁡Pj,whereA={j:d(Vi1,Bj)≤ε}.

Since the bi-objective optimization problem may have multiple solutions, the user can choose from its corresponding Pareto set a solution depending on whether total distance or total population coverage is prioritized. If the user prefers a single solution that incorporates both objectives, then we recommend that the optimization problem in Eq. (4) is considered.

Note that both optimization problems in Eqs. (4) and (5) assume that same sites are used throughout the vaccination program. To make the approach more dynamic, we propose a third optimization problem based on Eq. (4) that moves the vaccination sites towards the barangays which have yet to complete their vaccination program. Suppose there are S vaccination schedules. For k=1:S, we determine the optimal vaccination sites, denoted by Vi1(k),Vi2(k),…,ViL(k), during the kth schedule by solving

(6) min1≤i1,i2.…,iL≤M⁡∑j=1N⁡ηj(k)[PjTp+CjTc]min⁡{d(Vi1,Bj),d(Vi2,Bj),…,d(ViL,Bj)},

where ηj(k) is set to zero if the jth barangay has achieved the target percentage of vaccinated population during the kth schedule. Otherwise, ηj(k) is set to one. Note that the formulation in Eq. (6) is similar to Eq. (4) except for the indicator parameter ηj(k), which is introduced so that barangays who have completed their vaccination program will have no priority when choosing the optimal vaccination sites for the next schedule. In this study, we assume that the vaccination at the jth barangay is finished when 70% of its population has been vaccinated.

Numerical methods

Road distance using open street maps

For the overall numerical computation and some of the data extraction, we utilized the ease of use and availability of advanced open-source packages of the Python programming language. To compute for the driving or road distance between two points, we leverage Open Street Maps (OSM) and its corresponding Python package OSMNX. OSM is a dynamic repository of detailed map data such as road level data, buildings, and even natural geographic objects such as rivers and mountains. OSM is built and continues to be actively updated by contributors from diverse backgrounds such as hobbyist mappers, disaster risk experts, and GIS professionals. OSM is open source, which means anyone can access and use the full breadth of its data. OSMNX uses OSM data in conjunction with network graphs for a wide range of applications, such as all kinds of urban traffic and planning, all in a network graph analysis framework.

Single-objective optimization problem

In this subsection, we discuss the numerical algorithms that will be used to solve the single-objective minimization problems in Eqs. (4) and (6). To solve the optimization problem for a given municipality, the user must input two files: the village centers table and the vaccination centers table. The village centers table contains the number of COVID-19 cases, population, and location of all the barangays in the municipality. It is a CSV file with the schema given in Table 1. The vaccination centers table contains the location of all the possible vaccination sites. This is a CSV file with the schema shown in Table 2.

Table 1 The village centers table contains the number of COVID-19 cases, population, latitude, longitude, and names of all the villages or barangays in a town.

Infected
(data type: integer)	Population
(data type: integer)	Latitude
(data type: float)	Longitude
(data type: float)	Barangay_name (data type: string)	
Number of COVID-19 cases in the village/barangay	Population of the village/barangay	Latitude of the village hall/ community center/barangay hall	Longitude the village hall/ community center/ barangay hall	Name of the village/barangay	

Table 2 The vaccination centers table contains the latitude, longitude, and names of all possible vaccination sites in a town.

Latitude
(data type: float)	Longitude
(data type: float)	Name
(data type: string)	
Latitude of the vaccination center	Longitude of the vaccination center	Name of the vaccination center	

We found that it is possible to automate the extraction of the latitude and longitude data for the vaccination centers table using OSMNX to a considerable extent. However, the OSMNX automation could not differentiate between public and private schools, thus necessitating some manual review. OSMNX automation can be used to generate an initial version of the vaccination centers table on which the end-users can then build on by adding or removing vaccination centers to be considered. Even though the automation is only partial, it will still significantly reduce the manual processing needed to obtain a sufficiently good vaccination centers table. On the other hand, for the village centers table, OSM could not identify the village centers or barangay halls so manual extraction of this data using Google Maps was needed. This means that we had to first identify which building served as the barangay hall and then determine its latitude and longitude via Google Maps. In some cases, the coordinates of the barangay centers given by Google Maps were inaccurate. Thus, we used Google Street View to locate the building based on its address and then use that location’s coordinates for the latitude and longitude data. Once the barangay hall was identified and its coordinates finalized, we used various government data repositories to identity the most recent population of the barangay along with its number of infected cases. This was done manually for every barangay in the municipality until we completed the village centers table. To summarize, the vaccination centers table can largely be automated using OSMNX extraction and then manually tweaked by domain experts or policymakers in that region. Meanwhile, the village centers table must be constructed by hand using both Google Maps and government statistics databases. The partial automation of the vaccination centers table is shown in the Github repository for this article (Cabanilla, 2022) along with the rest of the program.

The cost function is computed directly as shown in Eq. (4), where the road distance d(Vi,Bj) between the ith vaccination site and the jth barangay hall is computed via OSMNX in Python. For both the single and L-site optimization, we iterate through every possible combination of all the vaccination sites and barangays so that the resulting optimum is the global optimum.

The Python program we developed takes in the two tables previously mentioned and outputs the assignments of each barangay center to its optimal vaccination site as well as a ranked list of other suboptimal combinations of vaccination centers and their respective costs. Since it is already ordered by cost, the optimum would be in the first row.

The code and a tutorial for the implementation of the numerical optimization method are found in Cabanilla (2022). Sample CSV files of the inputs can also be downloaded from this repository. The users can simply modify the CSV files for easier implementation. Using the road distance matrix {d(Vi,Bj)}∈RM×N calculated via OSMNX in Python, a MATLAB version of this enumerative technique can be found in Enriquez (2022).

For smaller values of L, the enumerative approach presented above is sufficient so that the global solution is obtained. For higher values of L, identifying the best solution from all possible combinations can be computationally expensive. Hence, an efficient optimization algorithm is needed. Observe that the objective function in Eq. (4) is an integer nonlinear programming problem. In this study, we use a genetic algorithm (GA) capable of solving mixed integer optimization problems (Deep et al., 2009) to solve Eq. (4) for higher values of L. GA has been shown to be effective in solving a wide range of applications in science and engineering (Khosravian et al., 2021; Zhang, 2019; Yang, Gomez & Blackburn, 2020; Katoch, Chauhan & Kumar, 2021; Velasco et al., 2020; Caro, Mendoza & Mendoza, 2021; Jamilla, Mendoza & Mendoza, 2021). For ease of use and open accessibility, the GA we implement is from the geneticalgorithm Python package, which has options for integer programming. All the hyperparameter settings are set to default values except for the number of iterations, population size, and maximum number of iterations without improvements before stopping. Because GA is probabilistic, the result of one run may differ from another. Although capable of attaining global minimizers, the obtained solution can be local in some cases. Hence, we run the genetic algorithm 300 * L times and store the best solution among these runs. To account for the dimensionality of the problem, particularly for L≥7, we suggest increasing the number of runs. We set the population size to 20 * L * L and the maximum number of iterations to 50, based on experimentation. The code implementing this optimization method can also be found in the GitHub repository (Cabanilla, 2022). Alternatively, a MATLAB version of the program can be downloaded in Enriquez (2022).

Multi-objective optimization problem

In this subsection, we discuss the numerical algorithm used to solve the bi-objective minimization problem given in Eq. (5). The algorithm requires the number of COVID-19 cases per barangay, population of each barangay, and road distances {d(Vi,Bj)}∈RM×N between each vaccination center and barangay. The user must also enter L, the desired number of optimal vaccination sites. The algorithm finds all possible combinations of sites taken L at a time and computes for the cost values of each combination based on the two objective functions given in Eq. (5). We note that in our experiments, we set the radius ( ε) in the second objective function to 3,000 m. The cost values are then listed in a 2-column vector, say C.

To obtain the Pareto optimal set, we apply a bubble sorting method. We first sort the rows of C based on increasing values of the first column of C, that is, increasing values of F1. This consequently makes the first row of C a member of the Pareto optimal set. Then, the algorithm treats the F2-value of this first member as the current-best and goes through the rest of the values of the second column of C, that is, the values of F2. If the algorithm finds a lower F2-value than the current-best, its corresponding row will then become a member of the Pareto optimal set, and the current-best is updated. This is done until all the rows of C have been checked. The combinations of L having the function values in the final Pareto optimal set represent the optimal vaccination sites that solve the bi-objective minimization problem. MATLAB was used for the implementation of this algorithm and the codes are available in Enriquez (2022).

Results

To illustrate how our proposed method works, we find the optimal placement of vaccination sites in San Juan, a municipality in the province of Batangas, Philippines. San Juan is comprised of 42 barangays. A map detailing the location of the barangays in San Juan is shown in the Supplemental File. Hence, {Bj,j=1,2,…,42} contains the locations of the 42 barangay halls in San Juan. The latitudes and longitudes of the barangay halls were manually obtained from local government directories and Google maps. Hospitals in San Juan are listed as possible vaccination sites. Since face-to-face classes were suspended in the Philippines during the COVID-19 pandemic, public schools (elementary, high school, and college) are also listed as possible vaccination sites (Ranada, 2021). In January 2021, the Catholic Bishops’ Conference of the Philippines offered to transform churches in the country as COVID-19 vaccination sites (Department of Health Press Release, 2021). The latitudes and longitudes of the hospitals, schools, and churches are obtained from the Philippine Department of Health and Department of Education directories, and Google maps. A total of 65 sites were identified in San Juan, consisting of five hospitals, 42 elementary schools, 13 junior high schools, two senior high schools, two universities, and one church. Hence, {Vi,i=1,2,…,65} contains the location of all the 65 possible vaccination sites. In cases when these sites are not available, one can easily modify the input to include other sites and exclude unavailable sites.

San Juan, Batangas has a projected population of 125, 252 in 2021 (Department of Health Publications, 2020). Meanwhile, as of May 31, 2021, San Juan recorded a total number of 579 confirmed COVID-19 cases. We chose May 31, 2021 because during this time, the vaccination program in San Juan, Batangas had just started. The complete information on the locations of possible vaccination sites and barangay halls in San Juan, the number of COVID-19 confirmed cases per barangay, and the population of San Juan per barangay are found in the Supplemental File.

Two outputs are provided by the codes. First, a geographic map of the area with the locations of the vaccination sites and barangay halls. Second, a data frame showing the vaccination site assignments of each barangay, as well as the distance between them. These results can be easily exported as a csv, excel, or any other format the user prefers.

For a sample implementation, we consider selecting one to four vaccination sites among {Vi,i=1,2,…,65}, that is, L=1,2,3,or 4. Figures 1 and 2 show the geographic distribution of the optimal vaccination sites in San Juan, Batangas along with their corresponding assigned barangays for L= 1 and 2 sites, and L=3 and 4 sites, respectively. The stars represent the optimal vaccination sites while the circular nodes are the barangay halls. All barangays assigned to a particular vaccination site have the same color. On the other hand, Fig. 3 illustrates a sample data frame output of the vaccination centers for ten barangays in San Juan assuming that there are only two vaccination sites. For instance, barangay ‘Abung’ is assigned to the vaccination site named ‘San Juan Rural Health Unit 1’. The distance between the barangay and the assigned vaccination site is 6,692.14 m. Similarly, barangay ‘Barualte’ is assigned to the vaccination site ‘Paaralang Elementarya ng Bataan’ and the distance between them is 2,693.79 m. Observe that the distance between barangay ‘Bataan’ and its assigned vaccination site is zero because the barangay hall of Barualte and the elementary school of Bataan are in the same compound.

Figure 1 The roadmap of San Juan, Batangas showing the optimal locations of one (left) or two (right) vaccination sites.

The dots represent the village/barangay halls while the stars are the computed optimal vaccination sites. The colors depict the vaccination site assignment of each village/barangay in the town.

Figure 2 The roadmap of San Juan, Batangas showing the optimal locations of three (left) or four (right) vaccination sites.

The dots represent the village/barangay halls while the stars are the computed optimal vaccination sites. The colors depict the vaccination site assignment of each village/barangay in the town.

Figure 3 Sample output of the algorithm showing ten barangays in San Juan, Batangas and the assigned vaccination site based on proximity.

The road distance (in meters) between the barangay and the assigned optimal vaccination site is also shown. Here, we assume that there are only two vaccination sites (L = 2).

Figure 4 shows the average distance (in kilometers) of the barangays in San Juan, Batangas to the assigned optimal vaccination site, for L=1,2,3, or 4 sites. Figure 5 displays the number of weeks it takes to vaccinate 70% of the population of San Juan for L=1,2,3,4,or5 sites, given different daily vaccination rates (100, 200, or 400 people per day).

Figure 4 Average road distance (in kilometers) of the barangays in San Juan, Batangas to the obtained optimal vaccination site for L = 1, 2, 3, or 4 sites.

Figure 5 The time needed to inoculate the first dose of COVID-19 vaccines to 70% of the population of San Juan, Batangas for L = 1, 2, 3, 4, or 5 sites, given a daily vaccination rate of 200 (orange), 400 (violet), or 100 (blue).

As mentioned earlier, the enumerative approach is used only for smaller values of L due to the limited memory capacity. For higher values of L, the problem was solved using GA. Table 3 displays the indices of the optimal vaccination sites for L=1,2,…,7. We observe that we obtain the same optimal sites for L=1,2,…,6 using GA and the enumerative approach. For L=7, the computer runs out of memory in generating all the possible combinations in the enumerative approach and no solution was obtained. On the other hand, GA was able to generate an optimal solution.

Table 3 Summary of results for the nonlinear integer programming using genetic algorithm compared with the enumerative approach for L = 1, 2,…7 sites.

Number of vaccination sites L	Optimal index/indices using genetic algorithm	Optimal index/indices using enumerative approach	
1	54	54	
2	[3, 9]	[3, 9]	
3	[1, 3, 24], [1, 3, 52]	[1, 3, 24], [1, 3, 52]	
4	[ 1, 2, 24, 59], [1, 2, 52, 59]	[ 1, 2, 24, 59], [1, 2, 52, 59]	
5	[1, 2, 15, 52, 59]	[1, 2, 15, 24, 59], [1, 2, 15, 52, 59]	
6	[1, 12, 15, 24, 30, 33]	[1, 12, 15, 24, 30, 33], [1, 12, 15, 30, 33, 52]	
7	[1, 9, 12, 14, 30, 51, 62]	Not solvable	

The results of the bi-objective optimization problem in Eq. (5) are presented in Figs. 6 and 7. The cost values of all site combinations for L=2, along with the corresponding Pareto optimal set (blue) and the optimal solution from the single-objective enumerative problem (red star) are illustrated in Fig. 6. All the other possible combinations of the vaccination sites are shown as green circles. Figure 7 shows the Pareto optimal sets for L=2,3,4, and 5.

Figure 6 Cost function values and Pareto optimal set of the bi-objective optimization problem for L = 2 sites.

The optimal solution of the single-objective (enumerative) optimization problem is also shown as a red star.

Figure 7 Pareto optimal sets of the bi-objective optimization problem for L = 2, 3, 4, or 5 sites.

The optimal solutions for each L of the single-objective (enumerative) optimization problem are also shown. Note that for L = 3, 4, 5, two vaccination site combinations obtained the optimal value, so they are represented as an overlapping star and diamond.

Figure 8 illustrates a sample result of the dynamic optimization approach in Eq. (6) assuming L=2 and a daily vaccination rate of 200. Figures 8A–8F demonstrate the monthly change in the locations of the optimal vaccination sites as more people were vaccinated. To simulate the vaccination process, random sampling was done to assign the barangay where the vaccinated individuals belong to and identify the remaining number of unvaccinated individuals in a barangay which is needed in the recalculation of the optimal sites. The sampling assumes that individuals residing in barangays close to the vaccination sites have higher probability of getting vaccinated. If a barangay has completed its vaccination program, that is, 70% of the population has been vaccinated, then it is excluded from the sampling. Figure 8A shows that the vaccination sites in the first month of the vaccination program are situated at ‘San Juan Rural Unit I’ (located in ‘Poblacion’, which is San Juan’s central barangay) and ‘Paaralang Elementarya ng Bataan’ (located in the barangay of ‘Bataan’). In Figs. 8A–8E, we observe that one of the vaccination sites did not change until after 4 months. This may be because this area (‘Poblacion’) contains the most populous barangays in San Juan and hence, the vaccination program here is expected to take time. In Fig. 8E, a site moved back to the south at the ‘Laiya Aplaya National High School’ to vaccinate the remaining residents of ‘Laiya Aplaya’. Figure 8F shows that the two vaccination sites moved to the northern part of the municipality, and the target population to be vaccinated has been completed.

Figure 8 (A–F) Solutions of the dynamic optimization approach for 6 months.

The vaccination sites are relocated after 1 month to move them closer to villages which have not completed their vaccination program. The red stars indicate the optimal vaccination sites while the circular nodes denote the location of the villages. A circular node is marked white when the vaccination program is finished. Otherwise, it is marked black.

Discussions

For the single-objective optimization problem, we observed that for L = 1, the optimal site location is close to the most populous area, which is in the northern part of the municipality. For L = 2, one optimal site is in the north (yellow star) and the other optimal site is in the south (purple star). The barangays assigned to the vaccination site in the north are represented by yellow dots, while the barangays assigned to the vaccination site in the south are represented by purple dots. As expected, the vaccination sites become more spaced out as the number of sites increases. In all cases, the optimal locations obtained are situated along the national highway since the problem is formulated to minimize the driving distance from the barangay halls to the sites. Notice that for L = 4, two vaccination sites out of the optimal three-site solutions did not change. The site in the northern part of the municipality was replaced by two sites. This is expected because this region is the most populated and has the greatest number of confirmed COVID-19 cases (see the Supplemental File).

On average, the difference between the road distance for one and two sites is approximately three kilometers while the difference between three and four sites is 600 m. The trend shows that as more vaccination sites are opened, accessibility to the vaccines, in terms of distance, is improved. However, opening more sites has associated operational costs. Results in Fig. 4 can provide information for the policymaker on finding a balance between accessibility and cost-effectiveness related to the number of vaccination sites to open.

Assuming a constant vaccination rate, we can determine the number of weeks it takes for a municipality to reach a target number of people to be vaccinated, say 70% of the total population (see Fig. 5). Suppose a site in San Juan can inoculate 200 individuals per day. This rate is based on the vaccination rate of the University of the Philippines Diliman gym in Quezon City (Ayalin, 2021). If there is only one vaccination site, it takes around 62 weeks to inoculate 70% of the population in San Juan. Meanwhile, increasing the number of sites to two shortens the number of weeks to 44. Observe that the difference in time between three and four sites is only 7 weeks. If the local government has the capacity to hold vaccinations at three sites only and wishes to achieve the target of vaccinating 70% of the population in 21 weeks (same length of time as in four sites), then the vaccination rate at the three sites can be ramped up by 34.5% or by vaccinating additional 69 people per day in the three sites.

If the municipality intends to identify a large number of vaccination sites, GA can be used. This can be useful for big municipalities or small cities. We have shown that if GA is given enough number of runs, the solution obtained can be the same as in the enumerative method.

For the bi-objective minimization problem, the Pareto optimal set is located at the lower left corner of Fig. 6. The Pareto set for L=2 contains five combinations of sites that minimize the distance to the barangays and maximize the population within ε-distance from the sites. The users are free to choose which combination of sites they prefer. If they prioritize proximity over population, then they may choose the sites with lower y-values. Otherwise, they can choose the combination with lower x-values. Observe also that the single-objective optimal solution for L≤4 is a member of the Pareto optimal set, which confirms that the single and bi-objective problems are consistent with each other. In Fig. 7, as L increases, the Pareto sets move further to the lower left area of the figure. This is expected because having more vaccination sites brings the sites closer to the barangays. That is, the total distance of the sites to the barangays is reduced and the population covered by the sites within a fixed radius is increased. While this is not the case for L=5, the single-objective solution is still close to the Pareto set. If one wishes to see that the optimal solution for L=5 is in the Pareto set, then one can vary the radius ε of the site to the barangays.

The dynamic optimization approach shows how the proposed scheme can be modified when the number of vaccination sites changes during the program. In this way, the vaccination sites can be relocated after a certain amount of time so that barangays which have not completed their vaccination program can gain more access to the vaccines.

In this study, we only considered minimizing the distance travelled and maximizing the population coverage because the scale of the study is small, and our main goal is to make the vaccination sites more accessible. This study does not consider other costs associated to vaccine delivery such as cold chain storage, waste management, transportation expenses, and technical assistance. Other factors which are not included in the costs can be due to coordination and planning, social mobilization, training of personnel, physician’s fee, and other miscellaneous costs (Siedner et al., 2022). We recognize that although these factors are important, these costs can be assumed to be the same for all the vaccination sites since the study is done at the municipal level. In this way, the costs will have no bearing in the formulated optimization problem. If the method is applied to a larger scale (provincial or national), then these costs may vary, and the problem should be reformulated. These are limitations of the study which may be pursued in future research.

Conclusions

In this study, we proposed an approach to strategically select COVID-19 vaccination sites from already existing facilities at the municipal level. In finding the optimal location of the COVID-19 vaccination sites, the method considers the location of the sites, population density of the municipality, and number of COVID-19 cases. An open-access program has been created to make the results reproducible. The code only requires two files, one is a list of possible vaccination sites and the other is a list of the barangays. Our numerical simulations show the strategic placements of vaccination sites to urge the people to get vaccinated as soon as possible. The method can be beneficial to underdeveloped rural municipalities in developing countries, where public transportation is not reliable or in some cases, not available.

Because the problem can be solved for a greater number of vaccination sites using GA, this approach can be extended not only to other municipalities, but also to big cities and provinces. Exploring other algorithms that can solve the proposed optimization problems is a research direction that can also be pursued. Moreover, one can extend the results of this study to find the optimal locations of new vaccination sites.

Although the method is intended for COVID-19 vaccinations, the method is general enough that it can be applied to formulating immunization or drug delivery strategies of other diseases. For example, if mass drug administration is to be implemented for school-age children for diseases like soil-transmitted helminths and schistosomiasis, then the locations can be restricted to just the elementary schools.

We hope that this study can help stakeholders in planning strategies to end the COVID-19 pandemic, which has crippled the world economy and has affected the lives of millions of people worldwide.

Supplemental Information

Supplemental Information 1 Supplementary Figure and Tables.

Figure S1 map credit: Marvin A. Sinag, https://commons.wikimedia.org/wiki/File:Ph_map_of_san_juan_batangas.png

Click here for additional data file.

Additional Information and Declarations

Competing Interests

Author Contributions

Data Availability

The authors declare that they have no competing interests.

Kurt Izak Cabanilla conceived and designed the experiments, performed the experiments, analyzed the data, prepared figures and/or tables, authored or reviewed drafts of the article, and approved the final draft.

Erika Antonette T Enriquez conceived and designed the experiments, performed the experiments, analyzed the data, prepared figures and/or tables, authored or reviewed drafts of the article, and approved the final draft.

Arrianne Crystal Velasco conceived and designed the experiments, performed the experiments, analyzed the data, prepared figures and/or tables, authored or reviewed drafts of the article, and approved the final draft.

Victoria May P Mendoza conceived and designed the experiments, performed the experiments, analyzed the data, prepared figures and/or tables, authored or reviewed drafts of the article, and approved the final draft.

Renier Mendoza conceived and designed the experiments, performed the experiments, analyzed the data, prepared figures and/or tables, authored or reviewed drafts of the article, and approved the final draft.

The following information was supplied regarding data availability:

The data is available in the Supplemental File.

The codes are available at GitHub:

- Cabanilla KI. 2021. Covid-Site-Optimization. https://github.com/kurtizak/Covid-Site-Optimization

- Enriquez EA. 2022. COVID-Vaccination-Sites. https://github.com/ErikaAntonette/COVID-Vaccination-Sites

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
