# Peer review of "Optimal selection of COVID-19 vaccination sites in the Philippines at the municipal level"

_PeerJ, doi:10.7717/peerj.14151_

## Round 0.1 · original submission · Minor Revisions

Dear Author,

Please revise based on the reviewers' suggestions.

Reviewer 1 ·

Basic reporting

The overall manuscript writing is very clear and easy to read and understand. It is well professional scientific drafted manuscript. I'm really impressed by the manuscript writing part.

Literature, reference, and background study have been adequately done by the authors.

Figures, tables, and raw data are sufficient and professionally structured manuscript.

Hypotheses are very clear and results are relevant according to the purposed hypotheses.

However, there is minor modification required before publication.

Experimental design

The study is coming within the aims and scope of this journal.

Mainly, this piece of research is focused on the COVID-19 pandemic vaccination survey based on machine learning. The authors have asked correct and relevant scientific questions and addressed them properly.

The authors have rigorously investigated the gap with high technical and ethical standards and the possibility of a reproducible dataset is high in the future.

Methods have been described with sufficient details and authors posted all scripts in the public domain for use and I believe the data is replicable.

Validity of the findings

The piece of research is very impactful in this pandemic vaccination and this research could be implemented in the other part of the world. Additionally, this piece of research will help stack-holder for plan a future vaccination and combat other pandemics in the future.

As far as I believe all the data is highly robust, statistically sound, and correct.

The conclusion needs to revise. It seems too longer. So, I suggested to reduce the size and keep your conclusion nearby your study and outcome.

Additional comments

The manuscript entitled “Optimal selection of COVID-19 vaccination sites at the municipal level” by Kurt Izak C et.al is a very exciting piece of research on COVID-19 vaccination management and evaluation at a small scale. This research could be expanded on at a larger scale in the future. I would recommend publication after minor revision.

Comment 1: The title seems to be very general, more global, and not aligned with the study. The title should be near-by your study and find. Therefore, I shall recommend changing the title and please mention “Philippines”.

Comment 2: Please check the grammar and properly cited paper. Overall, an introduction is looking perfect and has good enough content.

Comment 3: How to optimize and combine the Google image and tabular for OSMNX input in the python library. This part is not very clear and hard to understand. Could you elaborate more on how we combined this stuff for input?

Comment 4: Authors have performed a Genetic Algorithm (GA) on the MATLAB platform. However, MATLAB is paid and not freely accessible. Can you implement the same GA freely available library such as python? If possible (not mandatory).
Comment 5: The conclusion part is so long in words please be strict with your finding and nearby your results. Please reduce it.

Overall paper is designed and written very well and impressive. I would suggest publication after minor revision of this part of the study.

Best,
Sugandh

Reviewer 2 ·

Basic reporting

In the current study, Cabanilla et al. proposed an approach to determine the optimal location of coronavirus disease (COVID-19) vaccination sites. The authors presented how to select optimal vaccination sites from existing facilities to make the vaccines more accessible to the public, which in turn accelerates the nation's recovery from this pandemic. The author's study focused on finding the optimal location of vaccination sites that will make the vaccines more accessible to the population of the town or city. Some of the advantages that I find helpful are :
1. The proposed algorithm used a bi-objective optimization technique that incorporates distance and population coverage so that the policymaker can choose whichever needs to be prioritized.
2. The algorithm utilizes the Open Street Maps (OSM) and its corresponding Python package OSMNX to calculate the actual road distance.
3. The method can be applied to formulating drug delivery strategies for other diseases.

One minor comment is that the authors should include the GitHub repository in the main text to be easily/widely accessible.

Experimental design

Methods are described with sufficient detail & information to replicate.

Validity of the findings

Conclusions are well stated, linked to the original research question.

Reviewer 3 ·

Basic reporting

1. Check for typographical errors. For example, in line 187.
2. It is better to present Figures 1 and 2 using tables.

Experimental design

1. Distance and population density of the barangays are important but other factors must be considered such as the capacity of the vaccination sites. For example, if only one vaccination site is considered, maybe a bigger vaccination site must be selected to accommodate more people.

2. Aside from population density and COVID-19 cases, are there other factors that must be considered? The utility of the models must be improved.

3. Now that public schools are reopening and if the goal is to use the model in a long term, maybe the authors can consider other vaccination sites such as barangay/village basketball courts.

Validity of the findings

1. In line 337, what operational costs are being pertained here? Is this also applicable if different sites are considered for each schedule without increasing the vaccination sites for each schedule?

2. In Figure 7 and in the paragraph that starts in line 341, is it correct that one of the assumptions here is an everyday vaccination? Since one of the goals is to bring the vaccines closer to the people, is there an advantage in utilizing the same vaccination site/s everyday instead of choosing different site/s for each schedule that are closer to each barangay (different barangay/s for each schedule)? If possible, maybe the vaccination schedule (for example 1x or 2x a week and so on) can also be assigned by the user because not all municipalities offer an everyday vaccination.

3. In line 343, again, it is important that the capacity of the vaccination sites is more accurate to have a more accurate estimate of the date that the target population will be vaccinated.

Additional comments

Overall, the topic is very interesting and relevant but there are more factors to be considered for the model to be realistic and applicable.

Reviewer 4 ·

Basic reporting

In the abstract, lines 14-15, the authors may choose "town" and "villages" instead, for simplicity. They can expound this in details along the paper.
Lines 22-24 can be silent here. If the PeerJ can have an appendix or supplementary documents section, it can be reorganized there.

On Keywords, lines 28-29, arranged it alphabetically.

On Introduction: line 42-43, cite some works defining herd immunity.
Consider revising the sentence in lines 72-77, 98-99, 128-129, obeying right citation in APA, e.g., In Buhat et al. (2020)....Note: Authors can check all their citations obeying APA guidelines.

In line 132, state the version of Python package, if applicable or maybe also just provide the citation.

Consider revising lines 284-289 by incorporating figures, labels, or tables along one paragraph. This kind of suggestion can also be applied to some paragraphs to lessen the number of pages.

Numbers from 0 - 9 can be spelled-out followed by a number, e.g. line 294: only two (2) vaccination sites.

Line 390-394: This can be separated from the discussion and put in the limitation of the study before this section.

Authors could include in their literature review about vaccine hesitancy, coldchaine facility/freezers among others as factor for vaccine rollout challenges.

Experimental design

The approach of using Open Street Maps is commendable.

In line 108, ..." the open software that we developed" is unclear in the paper. Maybe the authors could cite or provide a development website for the mentioned open-software.

The minimization problem formulation is commendable.

Validity of the findings

No comment

Additional comments

The reviewer commended the authors, who came up with a somewhat realistic and implementable vaccination strategy for the Philippines.

---

## Round 0.2 · accepted · Accept

I have read and checked manually. The authors have addressed all the reviewers' comments and the manuscript quality of the manuscript was improved and it is ready for production.